# Detection of intracranial hypertension in children using optical coherence tomography: a systematic review

Sohaib R Rufai [ORCID] ,[1,2,3] Michael Hisaund,[1] Noor ul Owase Jeelani,[2] Rebecca J McLean[1]

[1]University of Leicester Ulverscroft Eye Unit, Leicester Royal Infirmary, Leicester, UK
[2]UCL Great Ormond Street Institute of Child Health and Craniofacial Unit, Great Ormond Street Hospital for Children, London, UK
[3]Clinical and Academic Department of Ophthalmology, Great Ormond Street Hospital for Children, London, UK

**Correspondence to**
Dr Rebecca J McLean;
rjm19@leicester.ac.uk

## ABSTRACT

**Objectives** To evaluate the diagnostic capability of optical coherence tomography (OCT) in children aged under 18 years old with intracranial hypertension (IH).

**Design** Systematic review.

**Methods** We conducted a systematic review using the following platforms to search the keywords 'optical coherence tomography' and 'intracranial hypertension' from inception to 2 April 2020: Cochrane Central Register of Controlled Trials, EMBASE, MEDLINE, PubMed and Web of Science, without language restrictions. Our search returned 2729 records, screened by two independent screeners. Studies were graded according to the Oxford Centre for Evidence-Based Medicine and National Institutes of Health Quality Assessment Tool for observational studies.

**Results** Twenty-one studies were included. Conditions included craniosynostosis (n=354 patients), idiopathic IH (IIH; n=102), space-occupying lesion (SOL; n=42) and other pathology (n=29). OCT measures included optic nerve parameters, rim parameters (notably retinal nerve fibre layer thickness) and retinal parameters. Levels of evidence included 2b (n=13 studies), 3b (n=4) and 4 (n=4). Quality of 10 studies was fair and 11 poor. There was inconsistency in OCT parameters and reference measures studied, although OCT did demonstrate good diagnostic capability for IH in craniosynostosis, IIH and SOL.

**Conclusions** This systematic review identified various studies involving OCT to assist diagnosis and management of IH in children with craniosynostosis, IIH, SOL and other pathology, in conjunction with established clinical measures of intracranial pressure. However, no level 1 evidence was identified. Validating prospective studies are, therefore, required to determine optimal OCT parameters in this role and to develop formal clinical guidelines.

**PROSPERO registration number** CRD42019154254.

## INTRODUCTION

Intracranial hypertension (IH) was first described by Quincke in 1896 and remains a subject of major clinical importance.[1] IH affects between 0.63 and 0.71 per 100 000 children.[2 3] Unaddressed IH can inflict devastating sequelae including visual impairment, neurocognitive delay, disability and death.[4 5] Subacute pathology in children can

cause insidious IH which may pose deleterious effects on the brain and vision before clinical detection. Thus, prompt detection and timely intervention is key in preventing or limiting the sequelae of IH.

Assessment of intracranial pressure (ICP) in children is notoriously difficult. Direct intraparenchymal measurement represents the gold standard, but carries numerous disadvantages including the need for overnight hospital admission, general anaesthesia and significant risk.[6 7] An ideal surveillance method would be highly sensitive, specific, safe, highly reproducible, rapid, non-invasive and child friendly with the capability to record serial measurements. Existing measures fail to fully satisfy all these criteria and often yield equivocal results in young children, including fundus examination,[8] B-scan ocular ultrasound,[9 10] radiology[11] and visual evoked potentials.[12 13]

IH causes optic nerve and retinal changes, which can be detected and quantified using optical coherence tomography (OCT)—a non-invasive imaging method to acquire ultrahigh resolution cross-sectional images of the optic nerve and retina within seconds.[14]

OCT has been successfully used to study the normal and abnormal development of the optic nerve[15] and fovea[16] in children, plus various conditions associated with IH.[17–23] Here, we conducted a systematic review to assess the role of OCT in detecting IH in children.

## METHODS

This systematic review was conducted in accordance with Preferred Reporting Items for Systematic Reviews and Meta-Analyses (PRISMA) guidelines[24] and the Cochrane Handbook.[25] The protocol is registered on PROSPERO[26] and published in *BMJ Open*.[27]

### Eligibility criteria for considering studies for this review

Eligibility criteria were established a priori and included OCT studies of children (aged under 18 years) with IH. Level 4 evidence and above was included, as per the Oxford Centre for Evidence-based Medicine (CEBM).[28] Exclusion criteria were: (1) studies of adults aged 18 or over; (2) studies not pertaining to IH; (3) studies not using OCT and (4) case reports and expert opinion without critical appraisal.

### Search methods for identifying studies

Medical subject headings terms for 'optical coherence tomography' and 'intracranial hypertension' were entered into search platforms: Cochrane Central Register of Controlled Trials, MEDLINE, EMBASE, PubMed and Web of Science. Online supplemental appendix 1 contains full details of our search terms and strategy. EndNote V.X9 (Thomson Reuters, New York, New York, USA) was used to manage data. No date/language restrictions were stipulated.

### Study selection

A three-stage, independent screening process was employed by two screeners (SRR and RJM), involving eligibility screening of titles, abstracts and full papers. Screening questions are listed in online supplemental appendix 2.

### Data collection and quality assessment

The main outcome measure was the diagnostic capability for OCT in detecting IH, expressed as diagnostic accuracy or by appropriate statistical testing.

Secondary outcome measures were:
► Condition(s) associated with IH per study.
► OCT device(s) used.
► OCT success rate.
► Other surrogate estimates of ICP.
► ICP range determined as normal.

Our data extraction tool was adapted from the Cochrane Collaboration (online supplemental appendix 3).[29] Evidence levels were graded by two independent graders (SRR and MH) as per the Oxford CEBM[28] and the National Institutes of Health (NIH) Quality Assessment Tool for Observational Studies[30] was applied for

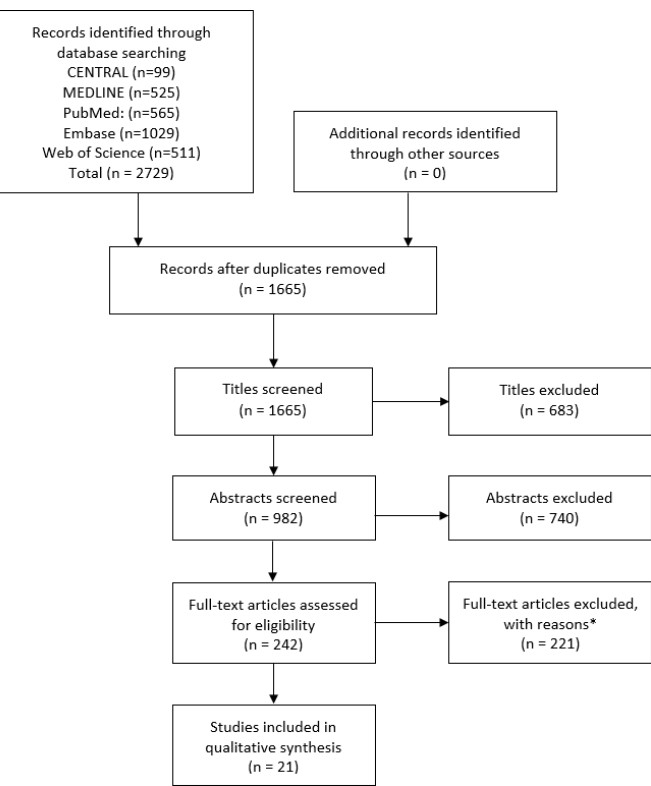

**Figure 1** PRISMA study inclusion flow diagram. *Reasons for exclusion were as follows: adult studies (n=98); mixed studies of adults and children without breakdown (n=27); conference abstracts (n=86); case reports (n=6); correspondence (n=4). Online supplemental table 1 contains the full list of excluded studies with reasons. CENTRAL, Cochrane Central Register of Controlled Trials; PRISMA, Preferred Reporting Items for Systematic Reviews and Meta-Analyses.

individual study quality grading (online supplemental appendix 4).

### Patient and public involvement

There was no patient and public involvement specific to this systematic review. However, this group has commenced prospective research using handheld OCT in paediatric IH featuring substantial patient and public involvement, which will be reported separately.

## RESULTS

Our search was executed on 2 April 2020 and data extraction completed on 12 July 2020. Our search returned 2729 records in total, 1665 following deduplication. Following full-text screening, 21 studies were eligible for inclusion in our review (figure 1). Online supplemental table 1 contains the list of excluded articles with reasons. Table 1 summarises the study characteristics and quality assessment of the 21 included studies. One study[31] was written in Polish and translated by an interpreter, while the remaining 20 were written in English. Nine studies were prospective[22 23 32–38] while 12 were retrospective.[31 39–49] No randomised controlled trials (RCTs) or

**Table 1** Study characteristics and quality ratings of included studies

| Source | Country | Study design | Conditions | Total number of study participants | Age | Level of evidence (1–5)* | Quality rating† |
|---|---|---|---|---|---|---|---|
| Prospective studies | | | | | | | |
| Chang et al[32] | USA | Prospective observational study | Other | 19 | M: 11; Ra: 5–17 | 2b | ⊕⊕⊕ |
| Den Ottelander et al[33] | The Netherlands | Prospective cohort study | Craniosynostosis | 38 | M: 0.7; Ra: 0.2–1.5 | 2b | ⊕⊕⊕ |
| Driessen et al[34] | The Netherlands | Prospective cohort study | Craniosynostosis | 38 | M: 6.2; Ra: 3.0–11.0 | 2b | ⊕⊕⊖ |
| El-Dairi et al[35] | USA | Prospective observational study | IIH | 48 | Mdn: 11; Ra: 4–14‡ | 2b | ⊕⊕⊕ |
| Lee et al[36] | USA | Prospective observational study | IIH | 13 | M: 14.8; SD 3.1‡ | 2b | ⊕⊕⊖ |
| Lee et al[37] | South Korea | Prospective observational study | SOL, other | 10 | M: 12.7; SD: 2.7 | 2b | ⊕⊕⊕ |
| Tran-Viet et al[38] | USA | Prospective feasibility study | Other | 11 | Ra: 0.7–2 | 3b | ⊕⊕⊕ |
| Swanson et al[22] | USA | Prospective observational study | Craniosynostosis, other | 79 | M: 2.9; SD: 3.8; Ra: 0.3–15.0‡ | 2b | ⊕⊕⊕ |
| Swanson et al[23] | USA | Prospective observational study | Craniosynostosis | 80 | M: 3.4; SD: 4.7; Ra: 0.2–18 | 2b | ⊕⊕⊖ |
| Retrospective studies | | | | | | | |
| Bialer et al[39] | Israel | Retrospective chart review | SOL | 20 | M: 6.5; SD: 3.9 | 2b | ⊕⊕⊖ |
| Dagi et al[40] | USA | Retrospective chart review | Craniosynostosis | 54 | Mdn: 9.3; SD: 4.6 | 2b | ⊕⊕⊕ |
| Dahlman-Noor et al[41] | UK | Retrospective cohort study | SOL, other | 61 | Mdn: 10.9; IQR: 7.9–13 | 2b | ⊕⊕⊕ |
| Gospe et al[42] | USA | Retrospective chart review | IIH | 31 | 7.8; SD: 3.4 | 2b | ⊕⊕⊕ |
| Malem et al[43] | UK | Case series | SOL, IIH, other | 20 | M: 11; Ra: 5–16 | 4 | ⊕⊕⊖ |
| Mediero et al[44] | Spain | Case series | SOL | 10 | Mdn: 5; Ra: 3–16 | 4 | ⊕⊕⊕ |
| Mrugacz et al[31] | Poland§ | Case series | IIH, other | 4 | Mdn: 9.5; Ra: 3–12 | 4 | ⊕⊕⊖ |
| Krishnakumar et al[45] | UK | Retrospective chart review | IIH | 16 | Mdn: 12; Ra: 3–15 | 3b | ⊕⊕⊕ |
| Ozturk et al[46] | Turkey | Retrospective observational study | IIH | 16 | M: 11.4; SD: 4.1; Ra: 3–17 | 3b | ⊕⊕⊖ |
| Sánchez-Tocino et al[47] | Spain | Case series | IIH | 3 | M: 8.3; SD: 3.8; Ra: 4–11 | 4 | ⊕⊕⊕ |
| Thompson et al[48] | USA | Retrospective cohort study | IIH | 90 | M: 12.2; SD: 3.7‡ | 2b | ⊕⊕⊕ |
| Van de Beeten et al[49] | The Netherlands | Retrospective chart review | Craniosynostosis | 104 | M: 0.9; Ra: 0.5–1.7 | 3b | ⊕⊕⊕ |

*Levels of evidence as per the Oxford Centre for Evidence-Based Medicine.
†The National Institutes of Health Quality Assessment Tool was used: ⊕⊕⊕, good; ⊕⊕⊖, fair; ⊕⊖⊖, poor.
‡Where separate age ranges were reported for patients and controls, the age range for patients is displayed here.
§Article written in Polish.
IIH, idiopathic intracranial hypertension; M, mean; Mdn, median; Ra, range; SOL, space-occupying lesion.

systematic reviews were eligible for inclusion. Following review of the included studies, it was deemed inappropriate to perform quantitative synthesis due to inconsistency in study design and methodology, particularly OCT parameters and reference standards. Therefore, qualitative synthesis was performed.

This review identified studies of craniosynostosis (n=354 patients), idiopathic IH (IIH; n=102), space-occupying lesion (SOL; n=42) and other pathologies (n=29) associated with risk of IH. OCT measures in these studies included optic nerve parameters, rim parameters and retinal parameters. Main outcome measures are displayed in table 2. Secondary outcome measures are reported per condition.

## Evidence summary: craniosynostosis

Craniosynostosis is characterised by the premature, pathological fusion of one or more cranial sutures. This restriction in skull growth can cause IH. Six studies in this review utilised OCT in patients with craniosynostosis.[22 23 33 34 40 49] These studies included a total of 393 participants, of which 354 were diagnosed with craniosynostosis.[22 23 33 34 40 49] In one comparative study,[22] 5 positive controls had hydrocephalus and suspected IH,[22] while 34 were normal controls.[22] OCT devices used include the iVue[22 23] (Optovue, Fremont, California, USA; software V.3.2) and Spectralis[33 34 40 49] (Heidelberg Engineering, Heidelberg, Germany).

OCT parameters evaluated include retinal nerve fibre layer (RNFL) thickness,[22 23 40] maximal retinal thickness[22 23 33 34 49] and anterior retinal projection.[22 23] OCT parameters demonstrated good diagnostic capability for IH in craniosynostosis (table 2). Increased RNFL thickness, maximal retinal thickness and anterior retinal projection were associated with papilloedema, while RNFL thinning was associated with optic atrophy. Using single, on-table ICP measures, Swanson et al[22] demonstrated good sensitivity (89%) and specificity (62%) of combined RNFL thickness and maximal anterior retinal projection in detecting IH (figure 2). They found that maximal RNFL thickness exceeding 207 μm or maximal anterior retinal projection exceeding 159 μm in either eye corresponded to the 97.5th percentile of healthy control patients, thereby representing IH; these figures did not vary significantly based on age.[22]

Reported OCT imaging success rates were high. Driessen et al[34] reported overall OCT success in 85% of eyes. Dagi et al[40] did not specify success rate, but reported that 16.9% were excluded due to limited cooperation, severe nystagmus, poor scan quality or retinal degeneration. Swanson et al implied 100% success rate in both studies,[22 23] but acquired OCT images under general anaesthesia and therefore were non-reliant on patient cooperation. Other studies of craniosynostosis did not report OCT success rates.

Apart from OCT, other surrogate measures of ICP displayed poor sensitivity, limiting their potential as screening tools for IH when used in isolation. These included fundoscopy,[22 23 33 40] visual acuity,[40] visual fields,[40] radiological signs[22 23 40] and clinical history including complaints of headache.[22 23]

With respect to defining IH, there is no universally agreed clinical consensus on timing, frequency and duration for accurate ICP measurement, or indeed what figure constitutes raised ICP.[50] In the three studies using ICP measurements,[22 33 49] ICP <10 mm Hg was determined as normal, while 10–15 mm Hg was determined as borderline and >15 mm Hg was determined as raised.

## Evidence summary: IIH

IIH, or primary IH, is characterised by raised ICP in the context of normal CSF composition and no evidence of SOL or ventriculomegaly on neuroimaging.[14] Principles from the modified Dandy criteria[51] and the revised criteria by Friedman et al[52] can assist in making diagnosis without ambiguity. Eight studies in this review utilised OCT in patients with IIH.[35 36 42 43 45–48] These studies included a total of 237 participants, of which 102 were diagnosed with IIH.[35 36 42 43 45–48] Of the other included participants in these studies, 74 had pseudopapilloedema,[43 46 48] 3 had SOL,[43] 6 had other pathology[43] covered below and 52 were normal controls.[35 46 48] OCT devices used include the Cirrus HD-OCT[14] (Carl Zeiss Meditec, Dublin, California, USA), Spectralis[42 48] and Stratus[35 47] (Carl Zeiss Meditec). Three studies did not specify which OCT device was used.[43 45 46]

OCT parameters included RNFL thickness,[35 36 42 47 48] macular volume,[35] disruption of the ellipsoid zone[42] and Bruch's membrane opening.[48] Two studies did not specify which OCT parameters were used.[45 46] OCT demonstrated good diagnostic capability in IIH (table 2). Increased RNFL thickness[35 36 43 48] and macular volume[35] were associated with IH, while RNFL thinning and disruption of the ellipsoid zone were associated with optic atrophy and vision loss.[42] In addition, Thompson et al[48] found that the transverse diameter of Bruch's membrane opening was enlarged in mild papilloedema and could be used together with RNFL thickness to distinguish mild papilloedema from psuedopapilloedema (figure 3).

Lee et al[36] reported 86.7% OCT imaging success rate, while Sánchez-Tocino et al[47] reported 100% OCT success. Other studies of IIH did not report success rates.

With regard to other surrogate measures of ICP, Ozturk et al[46] found that optic nerve sheath diameter was moderately associated with CSF opening pressure (r=0.661; p<0.005). Lee et al[36] found that body mass index was moderately associated with lumbar puncture opening pressure (r=0.607; p=0.028). Headache characteristics did not reliably detect children with IIH.[35 36 45–47]

With respect to ICP measurements, only Krishnakumar et al[45] provided definitions for normal and raised ICP, using lumbar CSF opening and steady state pressures:<15 mm Hg=normal CSF pressure;>20 mm Hg=high pressure.

**Table 2** Main outcome measures

| Source | OCT parameter(s) | Reference standard | Diagnostic capability of OCT |
|---|---|---|---|
| **Prospective studies** | | | |
| Chang et al[32] | (1) RNFL thickness; (2) volumetric SD-OCT of ONH; (3) volumetric EDI-OCT of ONH | Clinical diagnosis | (1) 69% accuracy; (2) 71% accuracy; (3) 67% accuracy |
| Den Ottelander et al[33] | TRT | Funduscopy and/or 24 hours ICP | No patients had abnormal TRT |
| Driessen et al[34] | TRT | Funduscopy | TRT increased in abnormal funduscopy (TRT p<0.001) |
| El-Dairi et al[35] | (1) RNFL thickness; (2) macular volume | Modified Dandy IIH criteria | (1) Thicker RNFL in IIH vs controls (p<0.0001); (2) thicker macular volume in IIH vs controls (p=0.0008) |
| Lee et al[36] | RNFL thickness | Funduscopy | RE: r=0.633, p=0.02; LE: r=0.868, p=0.001 |
| Lee et al[37] | (1) Neural canal diameter; (2) papillary vertical height; (3) anterior LC depth | Single ICP measures | (1) Postoperative reduction (p=0.027); (2) postoperative reduction (p<0.001); (3) postoperative increase (p=0.001) |
| Tran-Viet et al[38] | Not specified | Clinical diagnosis | Not assessed |
| Swanson et al[22] | (1) Maximal RNFL thickness; (2) maximal retinal thickness (3) maximal anterior retinal projection; (4) Combination of (1) and (3) | Single ICP measures | (1) Sens: 79%; Spec: 81%; (2) Sens: 63%; Spec: 86%; (3) Sens: 84%; Spec: 67%; (4) Sens: 89%; Spec: 62% |
| Swanson et al[23] | Combination of RNFL thickness, maximal retinal thickness and maximal anterior retinal projection | OCT | Not assessed |
| **Retrospective studies** | | | |
| Bialer et al[39] | RNFL thickness | Clinical diagnosis | Significantly lower in optic atrophy vs controls (p<0.001) |
| Dagi et al[40] | RNFL thickness | Funduscopy | Optic atrophy: Sens: 88%; spec: 94%; papilloedema: Sens: 60%; spec: 90%; all: Sens: 77%; spec: 83% |
| Dahlman-Nooret al[41] | (1) RNFL thickness; (2) anterior bowing of BM | Clinical diagnosis | (1) Present in two in three children; (2) present in one in three children. |
| Gospe et al[42] | (1) Optic atrophy: RNFL thickness <80 µm; (2) disruption of the ellipsoid zone (photoreceptor loss) | Revised Friedman IIH criteria | (1) OR for any vision loss: infinite (p<0.0001); (2) OR for any vision loss: 120 (p<0.0001) |
| Malem et al[43] | RNFL thickness | Neuroimaging±LP | Not assessed |
| Mediero et al[44] | (1) GCL thickness; (2) RNFL thickness | HVF | (1) kappa=1; p<0.001; (2) kappa=0.632; p=0.011 |
| Mrugacz et al[31] | RNFL thickness | Clinical diagnosis | Not assessed |
| Krishnakumar et al[45] | Not specified | LP | Not assessed |
| Ozturk et al[46] | Not specified | LP | Not assessed |
| Sánchez-Tocino et al[47] | RNFL thickness | LP | Not assessed |
| Thompson et al[48] | (1) BMO; (2) RNFL | Revised Friedman IIH criteria | (1) AUC=0.81; (2) AUC=0.96. Combined cut-offs: Sens: 91.7%; Spec: 92.2% |
| Van de Beeten et al[49] | TRT | Clinical diagnosis | Not assessed |

AUC, area under the curve; BMO, Bruch's membrane opening; EDI, enhanced depth imaging; GCL, ganglion celllayer; HVF, Humphrey visual fields; IIH, idiopathic intracranial hypertension; LC, lamina cribrosa; LE, left eye; LP, lumbar puncture; OCT, optical coherence tomography; op, operation; RE, right eye; RNFL, retinal nerve fibre layer; SD, spectral domain; TRT, total retinal thickness.

**A. Retinal thickness**

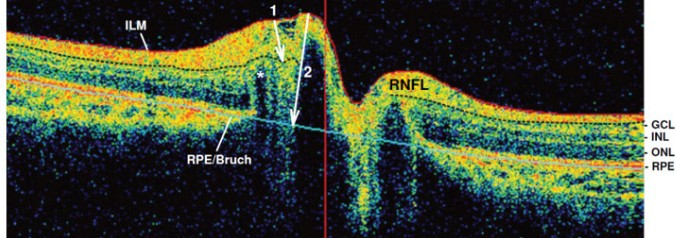

**B. Anterior retinal projection**

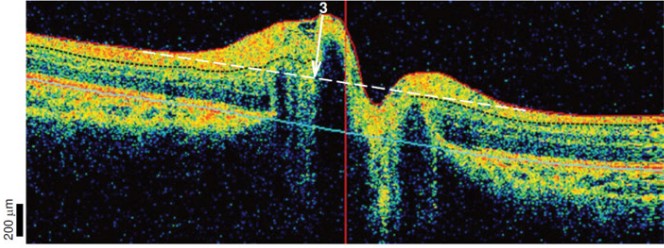

**C. Normal and elevated ICP**

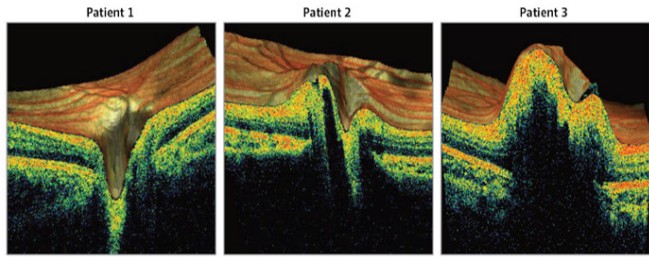

**D. OCT retinal parameters**

**E. ROC curves**

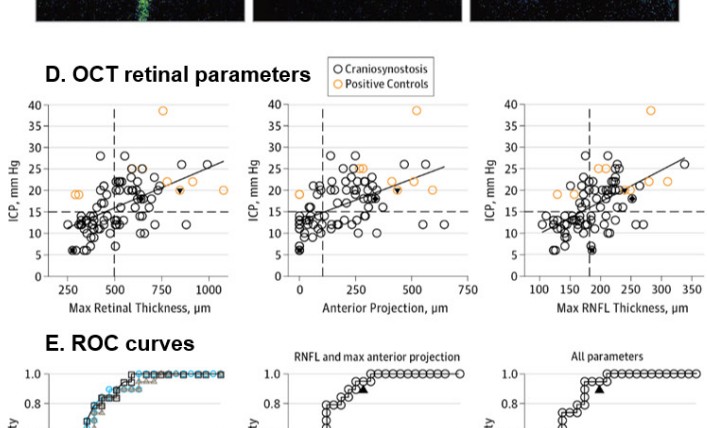

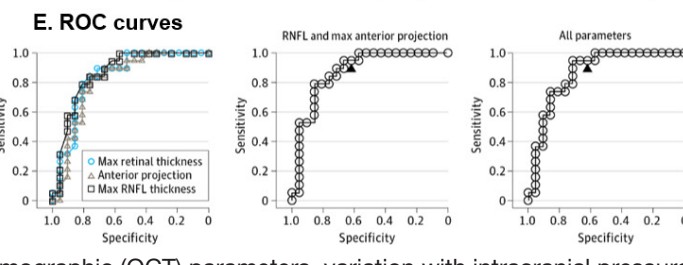

**Figure 2** Optical coherence tomographic (OCT) parameters, variation with intracranial pressure (ICP) and utility as a screening test. (A) 1. RNFL thickness; 2. Retinal thickness; *indicates vascular elements causing posterior shadowing; (B) 3. Anterior retinal projection, where the dotted white line is a vector connecting the posterior-most ILM adjacent to either side of the optic disc. (C) OCT images of patients with normal (patient 1; ICP, 6 mm Hg) and elevated (patient 2; ICP, 18 mm Hg and patient 3; ICP, 20 mm Hg) ICP. (D) OCT retinal parameters (maximal retinal thickness, anterior projection and maximal RNFL thickness) plotted as a function of ICP measured intraoperatively. (E) ROC curves for each of the three OCT parameters, combined RNFL and maximal retinal thickness parameters, and a model combining all parameters. GCL, ganglion cell layer; ILM, inner limiting membrane; INL, inner nuclear layer; ONL, outer nuclear layer; RNFL, retinal nerve fibre layer; ROC, receiver operating characteristic; RPE, retinal pigment epithelium. Reprinted with permission from: Swanson *et al*.[22] Copyright © 2017, American Medical Association.

### Evidence summary: SOL

Intracranial SOL include tumours or abscesses within the cranial cavity, which are associated with IH. Five studies in this review used OCT in patients with SOL.[37 39 41 43 44] These studies included a total of 121 participants, of which 42 had SOL.[37 39 41 43 44] Of the remaining participants, 2

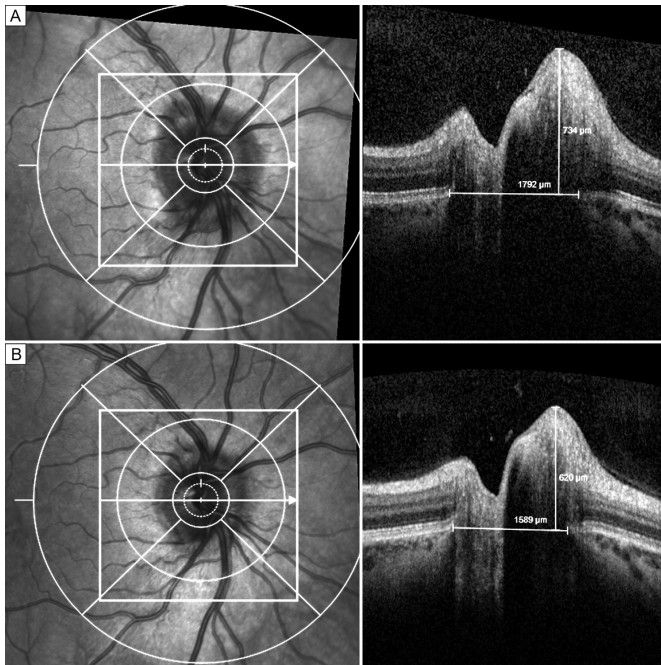

**Figure 3** Measuring the transverse horizontal diameter of Bruch's membrane opening (BMO) and the papillary height on SD-OCT. (A) When the ONH is swollen from mild papilloedema; (B) when the papilloedema has resolved. ONH, optic nerve head; SD-OCT, spectral domain optical coherence tomography. Reprinted with permission from: Thompson et al.[48] Copyright © 2017, Elsevier.

had IIH,[43] 9 had psuedopapilloedema,[43] 10 had other pathology[37 41 43] covered below and 58 had optic disc drusen with no intracranial pathology.[41] The following OCT devices were used: Cirrus HD-OCT,[39 44] DRI-OCT-1 Atlantis,[37] Spectralis[41] and Stratus.[39] Malem et al[43] did not specify which device was used.

OCT parameters included RNFL thickness,[39 41 43 44] ganglion cell layer thickness,[44] anterior bowing of Bruch's membrane,[41] neural canal diameter,[37] papillary vertical height[37] and anterior lamina cribrosa depth.[37] OCT demonstrated value as part of the clinical workup of these children (table 2). RNFL thinning was associated with optic atrophy[39] and visual field loss,[39 44] as was ganglion cell loss.[44] Lee et al[37] highlighted the reversibility of papilloedema in patients with SOL following ICP reducing surgery, whereby mean neural canal diameter and papillary vertical height decreased while mean anterior lamina cribrosa depth increased (figure 4). Dahlmann-Noor et al[41] reported one patient with intraventricular tumour with increased temporal RNFL thickness, but no bowing of Bruch's membrane.

Dahlmann-Noor et al[41] reported OCT imaging success rate of 100%, while other SOL studies did not report success rate.

Clinical history including headache characteristics did not reliably detect children with SOL.[39 41] Humphrey visual field testing is not designed for children and was difficult or not feasible in many cases.[39 44]

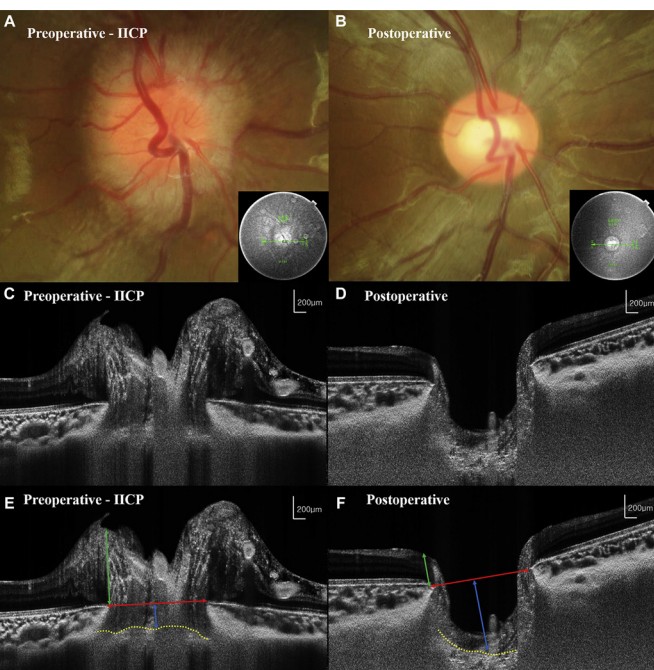

**Figure 4** Swept-source optical coherence tomography (SS-OCT) images (horizontal scan) of optic nerve head structures in the right eye of an 8-year-old boy with pilocytic astrocytoma. Disc photographs showing preoperative papilloedema (A) and postoperative resolution of the same (B) are presented with the indicating OCT section (small squares). A row SS-OCT image of preoperative (C) and postoperative (D) states can be observed. Measurements are obtained using the preoperative (E) and postoperative (F) images. After surgical decompression, the neural canal diameter (red line) and papillary vertical height (green line) have decreased and the lamina cribrosa (LC) shows posterior displacement (yellow dotted line; anterior surface of the LC contour). IICP, increased intracranial pressure. Reprinted with permission from: Lee et al.[37] Copyright © 2017, Elsevier.

Lee et al[37] reported mean preoperative and postoperative ICP values of 24.0±5.0 and 13.2±6.3 mm Hg, but no SOL study defined a normal range for ICP.[39 41]

### Evidence summary: other pathology

Seven studies[22 31 32 37 38 41 43] included children with other pathology associated with IH, including hydrocephalus[22 37 38] (n=16), papilloedema of unspecified aetiology[32] (n=5), acute lymphoblastic leukaemia[41] (n=1), aseptic meningitis[43] (n=1), growth hormone replacement therapy[31] (n=1), head injury[43] (n=1), recurrent nephrotic syndrome[31] (n=1), rickets[41] (n=1), sagittal sinus thrombosis[43] (n=3) and sigmoid sinus thrombosis[43] (n=1). Of note, Tran-Viet et al[38] used the Envisu handheld OCT system (C2200 and C2300, Bioptigen, Research Triangle Park, North Carolina, USA), with which they successfully scanned 25 out of 26 (96%) undilated eyes of conscious infants without sedation. Other OCT devices used were the Spectralis and DRI-OCT-1 Atlantis; Malem et al[43] did not specify which device was used.

OCT parameters used in these studies included RNFL thickness,[31 32 41 43] volumetric spectral domain

(SD) OCT of optic nerve head,[32] volumetric enhanced depth imaging (EDI) OCT of optic nerve head,[32] neural canal diameter,[37] papillary vertical height,[37] anterior lamina cribrosa depth[37] and anterior bowing of Bruch's membrane[41] (table 2). Tran-Viet et al[38] did not specify which OCT parameters were used. Chang et al[32] found a positive association between optic nerve head volume on SD-OCT and EDI-OCT with papilloedema, but did not specify aetiology. Other associations between OCT parameters and IH have been described above.

OCT imaging success rates were high in these studies: Tran-Viet et al[38] reported 96% success, Chang et al[32] reported 98% and Dahlmann-Noor et al[41] reported 100%. Mrugacz et al[31] and Swanson et al[22 23] implied 100% success rates. Other studies did not report OCT success rates.

## DISCUSSION

This systematic review collated a body of evidence evaluating the role of OCT to detect IH, specifically the structural changes associated with papilloedema and optic atrophy. This review could not recommend the widespread use of OCT in all children at risk of IH as standard clinical practice, as no level 1 studies were identified for these conditions. However, this review recognises the value of OCT in paediatric IH, particularly in cases where ICP status is uncertain or borderline, for a number of reasons: (1) OCT enables rapid, non-invasive, quantitative cross-sectional measurements of the optic nerve head and retina, not offered by conventional funduscopy; (2) OCT permits serial measurements to help appreciate evolution in optic nerve head and retinal changes over time, where applicable and (3) OCT could guide and support the overall clinical workup of affected children. Further prospective validating studies are required to develop formal clinical guidelines for OCT in this role.

### Quality of evidence

Two independent graders (SRR and MH) identified varying levels of evidence among the 21 included studies, as per the Oxford CEBM.[28] There were no systematic reviews, RCTs or validating cohort studies measuring OCT parameters against established reference standards, hence no study was graded as level 1 evidence. The NIH Quality Assessment Tool for individual studies identified 10 studies of fair quality and 11 studies of poor quality (table 1). The major limiting factor to quality was use of other surrogate clinical measures as the reference standard for IH, as opposed to direct intraparenchymal ICP monitoring.

### Research in context

There is a paucity of clinical guidelines for the management of paediatric IH in general. Searches for relevant clinical guidelines were performed via PubMed using the search terms "guidelines" AND "optical coherence tomography" AND "idiopathic intracranial hypertension" OR "craniosynostosis" OR "space occupying lesion", which

returned two relevant guidelines. Mollan et al[53] recently published the first consensus guidelines on management of IIH, which state: 'Where visual function is found to be threatened, regular ophthalmic examination must occur because this will influence timely management… Formal documentation of the optic nerve head appearance, such as serial photographs or OCT imaging, is useful.' This is consistent with our review's findings. The guidelines by Mollan et al[53] apply to all patients with IIH and are not specific to children. The Working Group on Craniosynostosis[54] published clinical guidelines for craniosynostosis in 2015, but these did not feature OCT. No relevant guidelines were identified for the use of OCT in SOL. The same search strategy was used for nice.org.uk—the UK's National Institute for Health and Care Excellence, which returned no relevant guidelines.

The NORDIC Idiopathic Intracranial Hypertension Treatment Trial is a landmark RCT with a dedicated OCT sub-study committee.[55] Although the resulting papers appeared in our systematic search, they were excluded from our review because they excluded children aged under 18. On further reading, positive associations between ICP and RNFL thickness, total retinal thickness and optic nerve head volume were also found, consistent with our review findings.[55]

This review identified studies using a range of OCT devices. Conventional, table-mounted OCT devices such as the Spectralis may be suitable in school-age children, but are not designed for young infants. While high OCT imaging success rates were reported, many studies were limited to school aged children rather than young infants. By contrast, this review found that the portable iVue device was successfully used for on-table OCTs in young infants under general anaesthesia,[22 23] while the Envisu handheld OCT was used in conscious newborns without the need for general anaesthesia or pupil dilation in one feasibility study.[38]

Of note, handheld OCT has been recently used to describe the normal development of the optic nerve[27] and fovea[15 16] in infants and children, with excellent feasibility. Handheld OCT has also been utilised in a wide range of other pedatric conditions including retinopathy of prematurity,[56] nystagmus,[57] albinism,[58] achromatopsia,[59] foveal hypoplasia,[60] optic nerve hypoplasia,[61] primary congenital glaucoma,[62] microcephaly[63] and others. Handheld OCT may be better tolerated in young children, particularly those with craniosynostosis associated with cognitive delay. However, further research is required to validate this. Therefore, our group has recently commenced prospective research using handheld OCT in craniosynostosis.[64]

### Strengths and limitations

This systematic review has a number of strengths. To the best of our knowledge, this is the first systematic review assessing the role of OCT in paediatric IH. Indeed, no such other systematic review appeared in our systematic search. PRISMA guidelines[24] and rigorous Cochrane

methods were followed.[25] Our protocol was registered on PROSPERO[26] and published[27] prior to this study, to promote transparency and avoid duplication. Two independent screeners (SRR and RJM) conducted the systematic search and two independent graders (SRR and MH) completed the quality assessment. A broad search strategy was developed with support from an experienced research librarian. Notably, our search terms did not restrict to certain conditions or age-related keywords—rather, the records were manually searched to avoid missing any key evidence. There were no time or language restrictions, yielding a broad range of eligible studies for inclusion including one article translated from Polish.[31]

We also acknowledge the limitations of this review. Meta-analysis was not possible due to the absence of eligible RCTs and inconsistency in OCT measures and reference standards. The maximum grade of recommendations from this review was grade B, based on consistent level 2 or 3 studies or extrapolations from level 1 studies.[28] Notably, while some studies demonstrated OCT detection of papilloedema (eg, RNFL thickening), other studies demonstrated OCT detection of optic atrophy (eg, RNFL thinning), therefore, caution must be exercised in interpreting OCT changes within the full clinical context and in serial examinations as far as possible. No level 1 evidence was returned by this review, which would be required to identify optimal OCT parameters and develop formal clinical guidelines. The broad search strategy resulted in a large number of records to screen, although this reduced the risk of missing key evidence.

### Further research

This review highlighted the lack of standardisation in OCT parameters used to detect IH in children. Further research is required to clarify the most appropriate OCT parameters for this purpose, using gold-standard ICP measures. This could be achieved by a validating prospective study using existing OCT reference standards. This should qualify as level 1b evidence as per the Oxford CEBM.[28] Handheld OCT could enable serial imaging in young infants without sedation, which would be particularly valuable as many pathologies associated with paediatric ICP can begin from birth, however current evidence is lacking. Serial OCT imaging could enable appreciation of evolving optic nerve head and retinal changes over time, where applicable. Further high quality prospective research could help to integrate OCT into formal clinical guidelines and clinical decision-making algorithms.

Other interesting research questions that were not answered in this systematic review include the following. First, how should OCT parameters be interpreted following IH where optic atrophy has occurred? The full clinical context, including visual function, may help indicate whether the patient has optic atrophy, rather than drawing conclusions based on OCT findings in isolation. Second, how should OCT parameters be interpreted in patients with gliosis of the optic nerve in chronic IH? This could prevent the typical ONH changes expected on OCT. Third, how should OCT be interpreted in recurrent IH which can occur in patients with ventriculoperitoneal or lumbar–peritoneal shunt obstruction? Again, gliosis may prevent the typical ONH changes expected on OCT. It is likely that OCT must be interpreted in the full clinical context, including visual function, to optimise diagnosis and management of these complex cases.

### CONCLUSIONS

This systematic review has recognised the diagnostic potential of OCT in paediatric IH in craniosynostosis, IIH, SOL and other pathology, in conjunction with established clinical measures of ICP, to guide diagnosis and management. However, this review could not recommend the development of formal guidelines, nor the widespread use of OCT in all children at risk of IH as standard clinical practice at this stage. Further validating prospective research is required to improve our understanding of the clinical utility of OCT in this role (including handheld OCT), to establish optimal OCT parameters for paediatric IH and to inform formal clinical guidelines.

**Acknowledgements** We thank the authors of the included studies for providing further data where required for our systematic review. We thank Dr Jordan Swanson, MD, for assisting with the adaptation of Figure 2 for this systematic review. We thank Selina Lock, Research Services Consultant at the University of Leicester David Wilson Library, for providing expert guidance in our systematic search strategy. We thank Krystyna Ulanicka for providing the English translation of the included Polish study.

**Contributors** Authorship and contribution: SRR: Design and conceptualisation of the study; major role in the acquisition of data; analysis and interpretation of the data; drafted the manuscript for intellectual content. MH: Analysis and interpretation of the data; revised the manuscript for intellectual content. NuOJ: Design and conceptualisation of the study; revised the manuscript for intellectual content. RJM: Guarantor; design and conceptualisation of the study; major role in the acquisition of data; analysis and interpretation of the data; revised the manuscript for intellectual content.

**Funding** SRR is funded by a National Institute for Health Research (NIHR) Doctoral Fellowship (Award ID: NIHR300155) for this research project. This publication presents independent research funded by the MRC and National Institute for Health Research (NIHR).

**Disclaimer** The views expressed are those of the author(s) and not necessarily those of the MRC, the NHS, the NIHR or the Department of Health and Social Care.

**Competing interests** None declared.

**Patient consent for publication** Not required.

**Ethics approval** No ethical approval was required to conduct this systematic review.

**Provenance and peer review** Not commissioned; externally peer reviewed.

**Data availability statement** No data are available.

**ORCID iD**

Sohaib R Rufai http://orcid.org/0000-0001-8134-6393

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
