## [Reviewer comments · BMJ Open]

ARTICLE DETAILS

TITLE (PROVISIONAL)	Detection of intracranial hypertension in children using optical coherence tomography: a systematic review
AUTHORS	Rufai, Sohaib; Hisaund, Michael; Jeelani, Noor ul Owase; McLean, Rebecca

VERSION 1 – REVIEW

REVIEWER	Banc, Ana Iuliu Hagieganu University of Medicine and Pharmacy Faculty of Medicine
REVIEW RETURNED	03-Mar-2021

GENERAL COMMENTS	The topic of this systematic review is of high clinical relevance, and the manuscript is very well written.
---

REVIEWER	Gama, Rita Hospital da Luz
REVIEW RETURNED	05-Mar-2021

GENERAL COMMENTS	page 6, lines 38-40: instead of "retinal thickness" use "maximal retinal thickness" page 7 line 28: photoreceptor loss" is a diagnose, not an OCT finding. Maybe "interruption of the ellipsoid" could be used. page 18: include a figure explaining what "anterior retinal projection" is, like the figure used in Swanson's article, reference number 42
--

REVIEWER	Elnahry, Ayman Cairo University
REVIEW RETURNED	30-Apr-2021

GENERAL COMMENTS	This study is well conducted and well written. Please define ICP in your abstract. The authors recommended the use of OCT in the diagnosis and management of pediatric IH. I agree with this recommendation, however, what is the authors' recommendation regarding the role of OCT in the follow up of IH? I believe this should be addressed separately since OCT may be misleading as an indirect indicator of IH during follow up. For example, cases that develop RNFL thinning due to optic atrophy during follow up may be mislabelled as improving if judged by OCT alone without functional tests. Also, gliosis of the optic nerve in chronic IH may prevent changes in the shape of the optic nerve with treatment. Also one important issue to address is the role of OCT in the detection of recurrent papilledema which may occur in patients with VP or LP shunt obstruction. What is the reliability of OCT in detecting these
--

	cases? Again, gliosis may prevent changes in the shape of the optic nerve. These questions may not have direct answers from the currently available studies but better to be mentioned and discussed further for comprehensiveness. In my experience for example, neither clinical examination nor OCT are always useful in detecting recurrent papilledema and visual function deterioration may be a better clue in detecting recurrence in some of these cases.
--	--

VERSION 1 – AUTHOR RESPONSE

Suggestion, Question, or Comment from Reviewer 1: Dr. Ana Banc	Authors' Response	Manuscript section featuring changes
The topic of this systematic review is of high clinical relevance, and the manuscript is very well written.	Many thanks to Dr Banc for her kind comments and positive feedback.	None requested.

Suggestion, Question, or Comment from Reviewer 2: Dr. Rita Gama	Authors' Response	Manuscript section featuring changes
page 6, lines 38-40: instead of "retinal thickness" use "maximal retinal thickness"	Many thanks – we have revised accordingly.	Results – Evidence summary: craniosynostosis, pg. 5
page 7 line 28: photoreceptor loss" is a diagnose, not an OCT finding. Maybe "interruption of the ellipsoid" could be used.	Many thanks – we agree and we have amended this to "disruption of the ellipsoid zone" as you have recommended.	Results – Evidence summary: idiopathic intracranial hypertension, pg. 6
page 18: include a figure explaining what "anterior retinal projection" is, like the figure used in Swanson's article, reference number 42	Many thanks for this suggestion – we have thus added the figure panels from the Swanson 2017 paper explaining anterior retinal projection, plus their other OCT parameters, incorporated into Figure 2. We have permission from Copyright Clearance Centre to reprint this figure from JAMA Ophthalmology.	Figure 2 revised

Suggestion, Question, or Comment from Reviewer 3: Dr. Ayman Elnahry	Authors' Response	Manuscript section featuring changes
---	-------------------	--------------------------------------

This study is well conducted and well written. Please define ICP in your abstract.	Many thanks for this positive feedback. As recommended, we have fully defined intracranial pressure in the abstract.	Abstract, pg. 2
The authors recommended the use of OCT in the diagnosis and management of pediatric IH. I agree with this recommendation, however, what is the authors' recommendation regarding the role of OCT in the follow up of IH? I believe this should be addressed separately since OCT may be misleading as an indirect indicator of IH during follow up. For example, cases that develop RNFL thinning due to optic atrophy during follow up may be mislabelled as improving if judged by OCT alone without functional tests. Also, gliosis of the optic nerve in chronic IH may prevent changes in the shape of the optic nerve with treatment. Also one important issue to address is the role of OCT in the detection of recurrent papilledema which may occur in patients with VP or LP shunt obstruction. What is the reliability of OCT in detecting these cases? Again, gliosis may prevent changes in the shape of the optic nerve. These questions may not have direct answers from the currently available studies but better to be mentioned and discussed further for comprehensiveness. In my experience for example, neither clinical examination nor OCT are always useful in detecting recurrent papilledema and visual	Many thanks to Dr. Elnahry for these excellent discussion points. We have included all these important questions in our additional paragraph in Further research (pg. 10): “Other interesting research questions that were not answered in this systematic review include the following. First, how should OCT parameters be interpreted following IH where optic atrophy has occurred? The full clinical context, including visual function, may help indicate whether the patient has optic atrophy, rather than drawing conclusions based on OCT findings in isolation. Second, how should OCT parameters be interpreted in patients with gliosis of the optic nerve in chronic IH? This could prevent the typical ONH changes expected on OCT. Third, how should OCT be interpreted in recurrent IH which can occur in patients with ventriculoperitoneal or lumbar–peritoneal shunt obstruction? Again, gliosis may prevent the typical ONH changes expected on OCT. It is likely that OCT must be interpreted in the full clinical context, including visual function, to optimise diagnosis and management of these complex cases.”	Discussion – Further research, pg. 10

function deterioration may be a better clue in detecting recurrence in some of these cases.		
---	--	--

VERSION 2 – REVIEW

REVIEWER	Elnahry, Ayman Cairo University
REVIEW RETURNED	02-Jul-2021
GENERAL COMMENTS	The authors performed all requested revisions.